# Light Absorption Enhancement and Laser-Induced Damage Ability Improvement of Aluminum Alloy 6061 with Non-Porous Alumina/CdSe@Al_2_O_3_/SiO_2_ Functional Gradient Films

**DOI:** 10.3390/nano12030559

**Published:** 2022-02-06

**Authors:** Jiaheng Yin, Lihua Lu, Yaowen Cui, Yongzhi Cao, Yunlong Du

**Affiliations:** Center for Precision Engineering, Harbin Institute of Technology, Harbin 150001, China; larryyin@hit.edu.cn (J.Y.); lihual@hit.edu.cn (L.L.); cflying@hit.edu.cn (Y.C.); ylhit@hit.edu.cn (Y.D.)

**Keywords:** CdSe QDs, AA 6061, light absorption, laser ablation

## Abstract

Numerical calculations of ultraviolet to near-infrared absorption spectra by cadmium selenide quantum dots (CdSe QDs) doped in anodic aluminum oxide pores were performed using a finite-difference time-domain model. The height, diameter, and periodic spacing of the pores were optimized. Light absorption by the dots was enhanced by increasing the height and decreasing the diameter of the pores. When the height was less than 1 μm, visible light absorption was enhanced as the spacing was reduced from 400 nm to 100 nm. No enhancement was observed for heights greater than 6 μm. Finally, the optical mode coupling of the aluminum oxide and the quantum dots was enhanced by decreasing the pore diameter and periodic spacing and increasing the height. Laser ablation verified light absorption enhancement by the CdSe QDs. The experiments verified the improvement in the laser-induced damage ability with a nanosecond laser at a wavelength of 355 nm after aluminum alloy 6061 was coated with functional films and fabricated based on numerical calculations.

## 1. Introduction

With the shortage of traditional fossil fuel resources such as coal, oil, and natural gas, coupled with the fact that they will cause serious environmental pollution and the greenhouse effect, finding new energy sources to replace traditional fossil fuels has become a major issue for contemporary technology to solve. For oil, the world’s total proven reserves exceed 1373 billion tons, and the reserve production ratio is about 43 years. For natural gas, the world’s total reserves are 141 trillion cubic meters, and the reserve production ratio is about 66 years. For coal, the world’s total reserves are 1043.8 billion tons. According to the current output, it is estimated that it can be mined for 235 years [1]. Among energy generation methods, the fusion energy produced by laser-driven, controlled inertial confinement fusion (ICF) is valued by all countries for its rich fuel, clean materials, and the safety of fusion reactors. Representative examples include the large-scale laser tritium–deuterium fusion device built at Lawrence Livermore National Laboratory (LLNL) and initiated by the U.S. Department of Energy in 1995, the “National Ignition Facility” (NIF) [2,3,4]. The new milestone reached in August 2021 was an energy yield from the target of more than 1.3 MJ, representing around 70 percent of the energy that the laser pulse had delivered to the fuel capsule in its sights, and “generating more than 10 quadrillion watts of fusion power for 100 trillionths of a second”, according to the NIF [5]. The French Atomic Energy and Alternative Energy Commission approved the “Laser Megajoule, LMJ” plan; the Rutherford–Alpton Laboratory of the Committee for Scientific and Technical Equipment of the United Kingdom used the Perwatt laser for the first time in the world; and the first Perwatt laser in Asia was built in Japan as the GEKKO-XII high-energy nanosecond at the Laser Engineering Institute of Osaka University. As part of the PEARL-X facility in Russia, there is a next-generation Perwatt laser facility. In China, an ultra-high-power laser facility at the Shanghai Institute of Optics and Mechanics also has world-leading equipment [6,7].

Aluminum alloy is used as the terminal optical component bracket due to its excellent mechanical properties. If stray light, ghost image, etc., are improperly handled, or due to the effect of the beam transmission system diaphragm, the laser will irradiate the surface of aluminum alloy and cause it damage [8,9]. In addition, the splashed metal particles may adhere to the surface of the optical element, causing secondary pollution on the surface of the optical element and reducing the laser-induced damage threshold (LIDT) of the optical element by about 60% [10,11,12,13]. Although we can continuously improve the processing and manufacturing accuracy of optical components, the influence of system stray light on the aluminum alloy “framework” is inevitable. Therefore, to alleviate the damage of optical components, stray light absorption management must be studied. While ensuring the processing quality of optical components, the study of laser-induced damage mechanisms and damage protection of aluminum alloy frames is also a key technical issue that cannot be ignored.

A key technical issue is the absorption of stray light [14] in nuclear energy systems. Aluminum alloy frames normally require surface treatment to eliminate stray reflections. Zhu et al. [15] fabricated a broadband plasmonic absorber with an average absorbance of ≈99% from 400 nm to 10 µm through assembly of Au nanoparticles onto a nano-porous template. In addition, Zhang et al. designed and illustrated an ultrathin Ag nanocomposite absorber with Ag nanocomposites, which could eliminate over 90% of stray light at 400–600 nm wavelengths. However, the low efficiency of preparation due to a complicated and time-consuming fabrication technology combined with the high expense of noble metals such as Au and Ag restrict its application in aluminum alloy frames. This requires light-absorbing materials in the pores of anodic aluminum oxide (AAO), which is a commonly used chemical conversion film formed on the surface of aluminum alloy. CdSe quantum dots (QDs) have been used to absorb light with relatively low cost [16,17]. Recently, Kohnehpoushi et al. [17] described visible light absorption enhancement in a CdSe-QD-sensitized TiO_2_ periodic nanorod array. The enhancement mechanism was related to the diameter, height, and periodic spacing of the TiO_2_ nanorods. Baffou et al. [18,19] extended the discrete dipole approximation and Green dyadic tensor method to simulate the thermodynamics of laser-irradiated plasma nanostructures, imaged the heat source distribution in light absorption systems (such as plasma nanostructures) by a molecular fluorescence anisotropy method, and verified the general physical rules controlling heat generation in plasma structures.

Here, we examined light absorption enhancement in AAO nanopores after the incorporation of CdSe QDs (see in Figure 1). In our design, CdSe QDs were filled in AAO pores, rather than in semiconductors such as TiO_2_, ZnO, or Si, because they can be used for eliminating stray light in ICF systems. Two-dimensional finite–difference time–domain (FDTD) calculations were used to solve Maxwell’s equations. The QD diameters were 10 nm and the grid size in the X- and Y-directions was 1 nm, while that in the Z-direction was 10^−4^ nm. Periodic boundary conditions were applied in the X-direction and perfectly matched boundary–layer conditions were used in the Y-direction. Ultraviolet (UV) to near-infrared light (200–1000 nm) in the p-polarization plane was incident in the forward Y-direction. In the FDTD simulations, each electric field component (Ex, Ey, Ez) was calculated at a different location within a Yee cell to determine the absorption profile, as given by Equation (1) as follows.
(1)$$P_{abs}=-0.5\omega {\left|E\right|}^{2}Im(\varepsilon_{\omega })$$
where *P_abs_* is the power absorbed per unit volume at each position, *ω* is the angular frequency, **E** is the total electric field amplitude, and *ε_ω_* is the permittivity of the material. They were generated by laser irradiation calculated by a particle counter which was firstly used to evaluate the LIDT of the materials. Nd: YAG laser was used to irradiate the material. The ablation craters are detected by scanning electron microscopy (SEM).

## 2. Theories and Simulation

Semiconductor nanocrystals (SNCs) such as cadmium selenide (CdSe) quantum dots offer wide applications in the fields of photovoltaics, solar energy harvesting, nanophotonics, imaging, sensing and other fields [20,21,22,23,24,25]. Since incident radiation causes excitation of free electrons, metal nanoparticles (MNPs) can generate intense electric fields in their vicinity [26]. When SNCs are held in the close vicinity of MNPs, the high electric fields induced by MNPs can lead to enhanced absorption in SNCs. These absorption phenomena in both nanostructures (MNPs and SNCs) are described by various theories. Plasmons originate from collective oscillations of free conduction electrons [27], while excitation is a bound state of electron–hole pairs [28], which makes the modeling of these hybrid hetero-structures very difficult, as to accurately describe the characteristics of the system, we need to effectively combine two different theories. The models which have been used to practically describe hetero-structure systems are complex. People regard nanostructures as independent units (applicable for a weak coupling regime) with independent dielectric functions which could be making classical electro-dynamic interactions within a finite–difference time–domain (FDTD) or a discrete dipole approximation (DDA) [29,30].

The numerical study of proposed functional films was conducted by using a commercial-grade simulator based on the FDTD method. (The basic idea of FDTD is to sample and discretize each electromagnetic field quantity E and H alternately in time and space. Each E(H) component is surrounded by four H(E) components. Using this discretization method, the Maxwell curl equation with time variables can be transformed into a set of difference equations that can be easily calculated by a computer) [31,32]. As long as the initial conditions, boundary conditions and material parameters are set, the electromagnetic field distribution of the whole space can be calculated step by step and iteratively on the time axis. In the process of FDTD calculation, the field component E(H) at a sampling point in space is associated with the surrounding field component H(E), and each medium parameter in the material equation is given to each Yee cell to reflect the role of the medium in the process of electromagnetic wave propagation. Therefore, this method can deal with the problems of target radiation and electromagnetic scattering with a non-uniform and complex shape and structure. Here, the nanostructure films were set as three functional films (nonporous alumina isolating film, a CdSe@Al_2_O_3_ nano-composites absorption film, and a SiO_2_ dielectric sealing protective film). Figure 1 shows the simulated unit cell of the periodic model. In the simulation procedure, the heights of the Al_2_O_3_ ellipsoids were 8 μm, 6 μm, 4 μm and 2 μm. The diameters of the nanopores were 60 nm, 70 nm, 80 nm and 100 nm with periodic spaces of 100 nm, 200 nm, 300 nm and 400 nm. Over the top layer, ultraviolet (UV) to near infrared (200–1000 nm) in the p-polarization plane was incident in the forward direction of the *y*-axis. Finally, the specific absorption was calculated using Equations (2) and (3). The dielectric function of materials used in the simulation was fitted with the experimental data to ensure that the simulated results agree with the measured ones.
(2)$$T_{\lambda }=\frac{\frac{1}{2}\int \mathrm{real}(P{\left(\lambda \right)}^{\mathrm{monitor}})\cdot dS}{\mathrm{sourcepower}}$$
(3)$$A_{\lambda }=1-T(\lambda )-R(\lambda )$$

Figure 2 plots the absorption spectra of AAO pores with embedded QDs for various height-to-diameter aspect ratios and periodic space. In Figure 2a, the light absorption was enhanced for wavelengths of less than 550 nm with increasing pore height and decreasing diameter. In the infrared region, decreased height and diameter enhanced light absorption. This was consistent with light absorption as a function of periodic pore spacing over the range 200–400 nm (Figure 2b–d). The results for 200 nm and 300 nm spaces were close, and the light absorption sharply decreased for a 400 nm spacing. Figure 2a–d revealed shifts in the absorption peak in the 300–500 nm range toward lower wavelengths with increasing pore spacing, which indicates an increase in the AAO bandgap. The enhanced light absorption was largely related to the height of the pores. Thus, in the following, the heights of pores embedded with QDs were increased to optimize absorption.

Figure 3 plots the light absorption enhancement in AAO pores with embedded QDs for 100 nm, 200 nm, 300 nm, and 400 nm spaces. To optimize the spacing and height, the pore diameter-to-height ratios were 60:8000, 70:6000, 80:4000, and 100:2000. In Figure 3a–d, visible light absorption was enhanced with increasing spacing over the range 100–300 nm. However, when the spacing was 400 nm, the light absorption began to decrease. In the infrared region, the absorption enhancement at the 100 nm spacing was far greater than the other spaces. When the diameter-to-height ratios were 60:8000 and 70:6000, the enhancement of the 100 nm spacing was more than 95% (Figure 3a,b). When the height was 2000 nm, the enhancement was about 90%, while those for other spaces were less than 60%. At the UV band, the absorption enhancement increased with increasing height and decreasing diameter. However, in Figure 3a, the enhancement was not apparent at heights over the range 6–8 μm.

## 3. Experiment Procedure

In our studies, AA 6061 worked as a substrate with a length, width and thickness of 10 mm, 25 mm, and 3 mm, respectively. The chemical composition of AA 6061 and laser parameters can be found in ref. [9]. An Nd:YAG laser operating at 355 nm was used (schematics of laser ablation testing equipment shown in Appendix A). The 355 nm laser pulse duration was 6 ns, the area of the gaussian laser spot was 0.7 mm^2^, and the repetition rate was 1 Hz. The instruments used in our experiments include an Nd:YAG laser, collimated light source, focusing lens, splitter wedges, an EPM2000 energy calorimeter, a sample carrier (two-dimensional, adjustable, step-by-step accuracy is 10 μm), an optical microscope and a computer for controlling and data acquisition. Laser ablation craters were studied with scanning electron microscopy (SEM), and a particle counter was used to record the number of particles created with various diameters. The laser device was placed with the level of organic pollutants from A/10 to A, which means the range of nonvolatile residues was 0.1 μg/cm^2^ to 1 μg/cm^2^. In addition, the level of particulate pollutants is between 50 and 300, which means the density of particles greater than 5 μm is between 166 and 170,000 per square foot.

All of the films were prepared via anodic oxidation. The process included degreasing, alkali corrosion, neutralization, anodization, electrolytic deposition, and sealing. The degreasing solution was composed of aqueous solutions of sodium salts or bases such as Na_2_SO_4_ and NaOH. The treatment was at 50–65 °C for 5–10 min. The alkali etching solution included NaOH at a concentration of 50–80 g/L and commercial additives. The bath temperature was 50–75 °C. The natural oxide film on the surface of the aluminum alloy was removed by soaking for 10–15 min. The surface was then immersed in a neutralizing solution with HNO_3_ as the main component, and the metallic luster was restored. The metal matrix was completely exposed and easily corroded. Thus, if oxidation and subsequent protection steps could not be performed immediately, it was soaked in deionized water to avoid contact with air. The anodization processes were conducted at 0 °C, 12 V and various times to acquire different film heights. CdSe nanoparticles were prepared inside the AAO pore. Commercial sealing reagent was applied for the sealing step.

## 4. Results and Discussion

To verify the accuracy of the simulations and to demonstrate that the surface treatment of the aluminum alloy 6061 (AA 6061) increases light absorption, spectra were acquired before and after surface treatment. The height of the absorption layer was about 8 μm and the pore diameters were about 60 nm. Cross sections of functional films are shown in Figure 4a. Figure 4b–f demonstrate the distribution of elements of functional films. The experimental results in Figure 4 and Figure 5 indicate that regardless of the surface quality or the generated particles following surface anodizing, the AAO pores with embedded QDs exhibited higher light absorption than undoped AA 6061. In most cases, more absorption indicated greater damage. However, the absorbed light was diffused in deeper regions and the SiO_2_ acted as a protective layer for the AA 6061 substrate. The melting and vaporization points of AAO are 2327 K and 3253 K, respectively. Those for SiO_2_ are 2273 K and 2973 K, and those for AA 6061 are 923 K and 2740 K, respectively. These differences indicated that the protective effects were better than the absorptive effects. Hence, the functional layers absorbed more and experienced less damage because of the AAO pores embedded with QDs. The QDs thus absorbed the light, and the higher absorption values of the AAO and the SiO_2_ films provided the AA 6061 with greater protection against laser-induced damage. The thickness of SiO_2_ film is about 200 nm, so the distribution of the surface is only the silicon element and oxygen element.

Figure 5 shows the surface morphology of AA 6061 after 15 laser pulses with a 480 μm spot radius. In Figure 5a, AA 6061 coated with a nonporous alumina isolating layer, a CdSe@Al_2_O_3_ nanocomposite absorption layer, and a SiO_2_ dielectric sealing protective layer, the radius of the heat zone was less than 100 μm, and there was a cluster of ablation pits after solidification. The ablation zone cannot be seen, which was significantly smaller than the 0.7 mm^2^ laser spot size. (The enlarged view of the ablation area is shown in Figure 5b). This indicates that the surface treatment with functional layers absorbed light and protected the AA 6061 substrate. However, Figure 5c shows the morphology after being exposed to 15-pulses of laser radiation of AA 6061. (The enlarged view of the ablation area is shown in Figure 5d). The areas of the heat zones were greater than 0.5 mm^2^, which indicated damage at the edge of the Gaussian laser spot at a low power density. The enlarged figure in the upper right corner depicts the loose-layered structure after cooling for several days. It was in an unstable state that could be peeled away. To further verify that the AAO pores with embedded QDs absorbed the light, we used a particle counter to record the number of particles with various diameters created by the laser irradiation. Greater light absorption indicated stronger protection of the upper layer membrane and less particle contamination.

Figure 6 shows particles generated by fifteen pulses of laser irradiation. (The specific number of particles is shown in Appendix A in the Appendix A). The particle numbers and their diameters were recorded after each laser shot. The total number of particles of AAO pores with embedded QDs was 1239. A total of 57.6% of the 0.3 μm-diameter particles were produced during the first three laser pulses, and nearly 70% of the rest were generated (Figure 6a). However, 31,300 particles were generated when the AA 6061 surface was not treated with AAO pores with embedded QDs. More than 90% of the various particle diameters were produced during the first three laser pulses. This indicated that, without treatment, the surface could be easily damaged by laser irradiation. There were 101 particles with diameters of 5 μm. These could significantly contaminate an ICF system and affect optical transmission (Figure 6b). 

Figure 7 demonstrates the morphologies of AA 6061 coated by AAO pores with embedded CdSe QDs and SiO_2_ composite films at 1064 nm/0.5 J/cm^2^ laser irradiation for (a) 1 time, (b) 5 times, (c) 10 times, (d) 20 times, (e) 50 times, and (f) 100 times. Comparing the morphologies of AA 6061 with functional films at a wavelength of 355 nm, the ablation areas and depths are more obvious. According to simulation results, the light absorption of 355 nm is about 90%, which is far more than that of 1064 nm wavelength. (Morphologies of 355 nm laser irradiation can be seen in Appendix A.) The energy levels of molecules mainly include the energy levels of electrons, the vibrational energy levels corresponding to the relative motion of atomic nuclei in molecules, and the rotational energy levels corresponding to the overall rotation of molecules. The interval between the vibrational energy level and the rotational energy level of most material molecules is the energy corresponding to the infrared photon. Therefore, when most substances are bathed in infrared light, material molecules can absorb a large number of infrared photons and transition to a high energy level with faster vibration or rotation. The acceleration of molecular vibration or rotation in the micro corresponds to the temperature rise of the macro object being heated. The energy of ultraviolet photons mostly corresponds to the energy level of electrons in the molecules mentioned above. However, a UV laser can often make electrons transition from one atom to another so as to change the structure of the whole molecule, that is, photochemical ablation.

## 5. Conclusions

In conclusion, enhanced light absorption in AAO pores with embedded CdSe QDs was investigated using FDTD-based simulations. Visible light absorption increased when the pore height was increased and the diameter was decreased. It was not enhanced when the height was greater than 8 μm. The absorption edge shifted toward UV wavelengths, which indicated an increased equivalent bandgap for the AAO pores with embedded QDs. In the near-infrared region, a large periodic spacing (*p* = 400 nm) would lead to a higher absorptivity for increased heights, relative to that observed for the 100 nm, 200 nm, and 300 nm spaces. Finally, experiments indicated that AAO pores with embedded QDs improved AA 6061 resistance to laser-induced damage at a wavelength of 355 nm. The Appendix A indicates the improvement of the anti-laser-induced ability of AA 6061, which could be explained as follows: firstly, the functional gradient films composited of non-porous alumina/CdSe@Al_2_O_3_/SiO_2_ have a good absorptivity to stray light. Then, the multi-films smooth the defects of the rough surface of AA 6061. Finally, the vaporization temperature of SiO_2_, which worked as protection layer, is 2973 K (more than that of AA 6061-2740 K) and improved its laser-induced damage threshold. The damage zone and ejected particles were much less pronounced than for untreated materials. The simulations agreed well with the experimental results and demonstrated that anodic aluminum oxide pores with embedded CdSe QDs enhanced the light absorption of AA 6061. Finally, the non-porous alumina/CdSe@Al_2_O_3_/SiO_2_ functional gradient films can effectively absorb 355 nm UV stray light and improve the anti-laser-induced ability of AA 6061.

## Figures and Tables

**Figure 1 nanomaterials-12-00559-f001:**
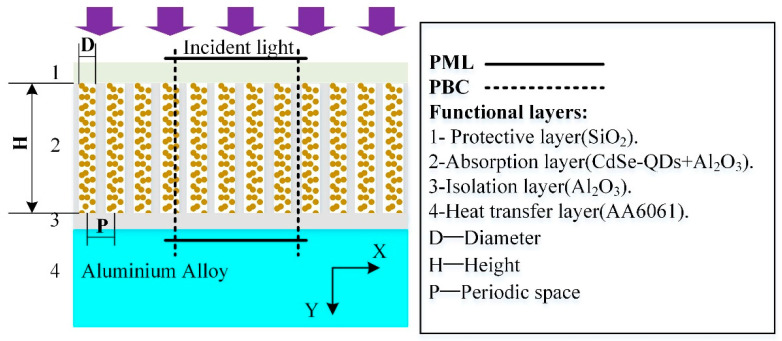
Schematic of simulated anodic aluminum oxide nanopores with embedded CdSe quantum dots.

**Figure 2 nanomaterials-12-00559-f002:**
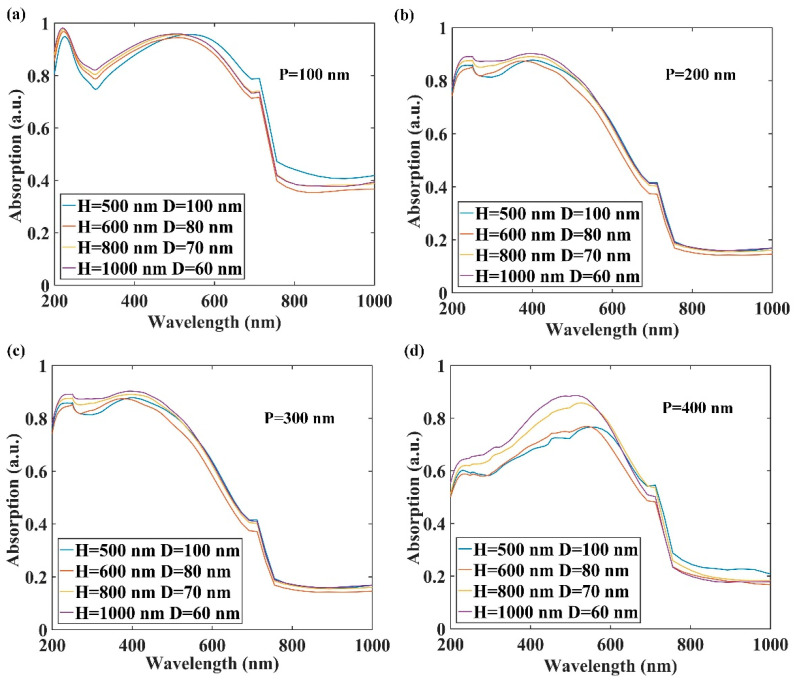
Light absorption enhancement in anodic aluminum oxide pores with embedded CdSe quantum dots at periodic spaces of (**a**) P = 100 nm, (**b**) P = 200 nm, (**c**) P = 300 nm, and (**d**) P = 400 nm.

**Figure 3 nanomaterials-12-00559-f003:**
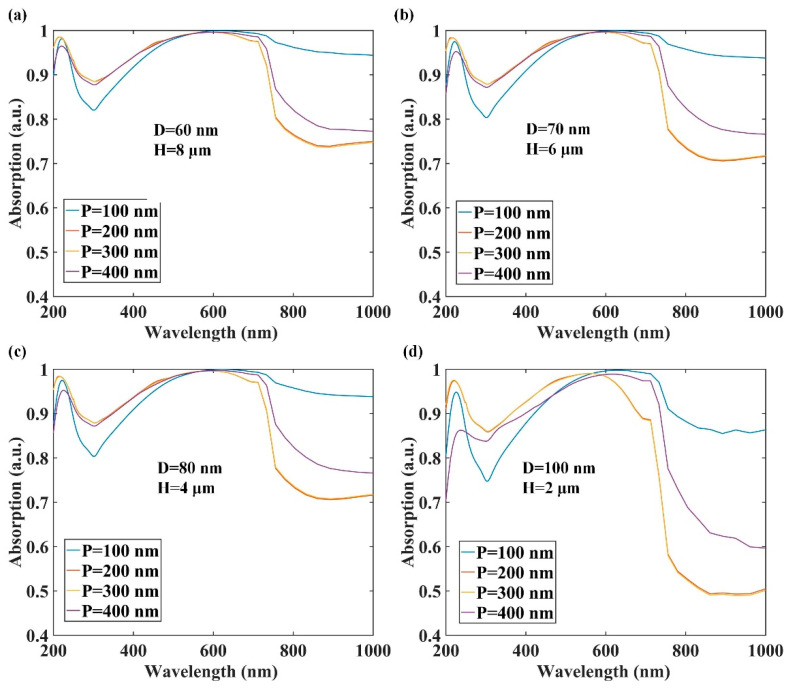
Light absorption enhancement in anodic aluminum oxide pores with embedded CdSe quantum dots and 100 nm, 200 nm, 300 nm, and 400 nm periodic spaces for (**a**) D = 60 nm and H = 8 μm, (**b**) D = 70 nm and H = 6 μm, (**c**) D = 80 nm and H = 4 μm, and (**d**) D = 100 nm and H = 2 μm.

**Figure 4 nanomaterials-12-00559-f004:**
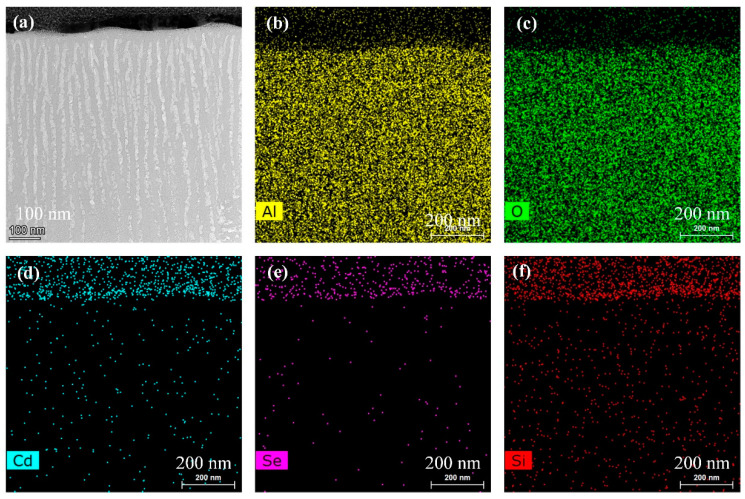
(**a**) Cross section of functional layers. Distribution of (**b**) aluminum, (**c**) oxygen, (**d**) cadmium, (**e**) selenium, and (**f**) silicon of functional layers.

**Figure 5 nanomaterials-12-00559-f005:**
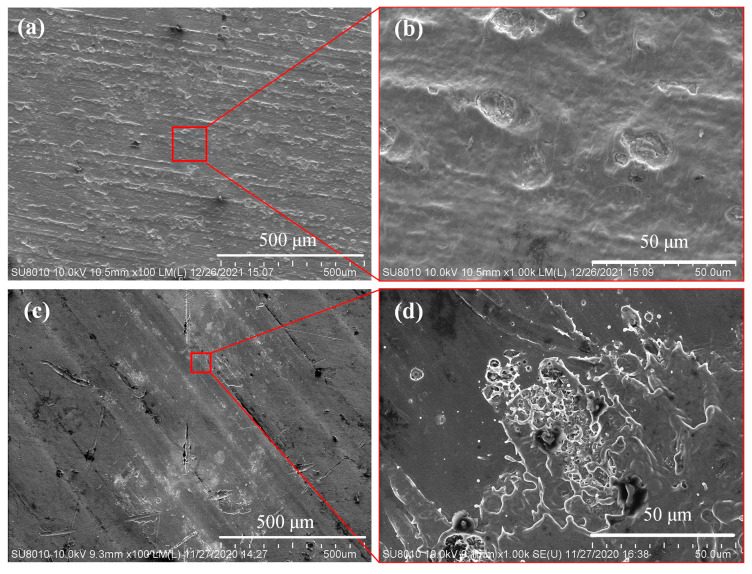
355-nm Nd:YAG laser irradiation of aluminum alloy 6061 at a fluence of 0.5 J/cm^2^ after laser irradiation of 15 times. (**a**) With functional layers. (**b**) Enlarged view of ablation area of Figure 4a. (**c**) Without functional layers. (**d**) Enlarged view of ablation area of Figure 4c.

**Figure 6 nanomaterials-12-00559-f006:**
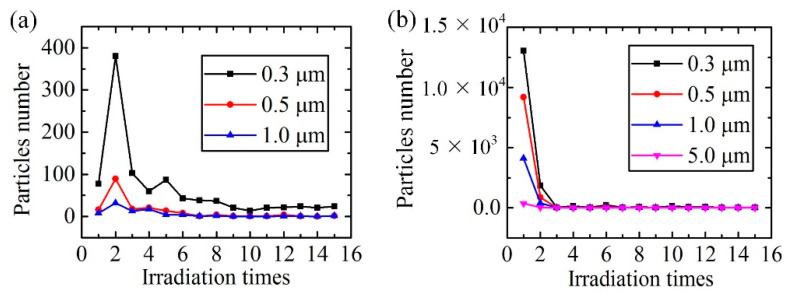
Number and diameters of particles produced by a laser fluence of 0.5 J/cm^2^. (**a**) With functional layers. (**b**) Without functional layers.

**Figure 7 nanomaterials-12-00559-f007:**
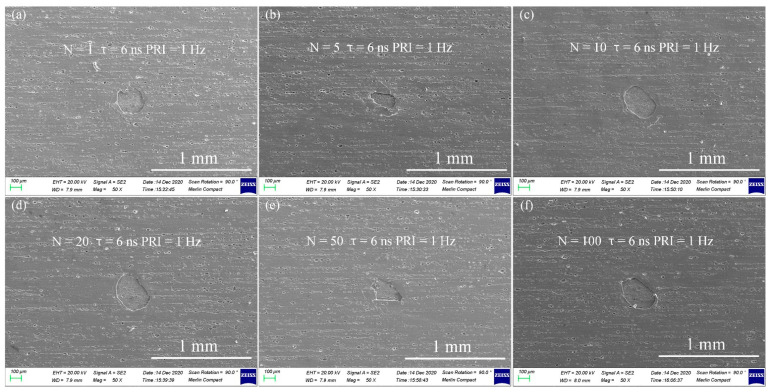
1064 nm Nd:YAG laser irradiation of aluminum alloy 6061 with functional layers at a fluence of 0.5 J/cm^2^ at (**a**) 1 time, (**b**) 5 time, (**c**) 10 time, (**d**) 20 time, (**e**) 50 time, (**f**) 100 time irradiation.

## Data Availability

Not applicable.

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
