# Peer review of "Light Absorption Enhancement and Laser-Induced Damage Ability Improvement of Aluminum Alloy 6061 with Non-Porous Alumina/CdSe@Al_2_O_3_/SiO_2_ Functional Gradient Films"

_nanomaterials, 2022, doi:10.3390/nano12030559_

Round 1
Reviewer 1 Report
The issue of the studying a light absorption enhancement and laser-induced damage ability improvement of different technical materials (which are used in fusion reactors in problems of laser-driven controlled inertial confinement fusion etc) has nowadays attracted very considerable attention. It is appropriate to recall the most significant progress in inertial thermonuclear fusion in 2021 in Russia and the United states since the start of research in 1972. Moreover, the desired progress is associated with a number of improvements, in particular, an increase in the size of the hohlraum and capsule, an increase in X-ray energy (which this capsule can absorb), an increase in the duration of the laser pulse, modification and replacement of capsules with denser ones in order to more efficiently compress the fuel. Also, some technical improvements included smoothing out microscopic irregularities on the surface of the fuel capsule, reducing the size of the hole in the capsule used for fuel injection, and reducing the holes in the cylinder so that less energy escapes.
The paper by Jiaheng Yin et al is devoted to studying a light absorption enhancement and laser-induced damage ability improvement of so called AA 6061 (?) with non-porous alumina CdSe@Al2O3/SiO2 functional gradient films. The authors presented the results of numerical calculations of absorption spectra (in the range from ultraviolet to near infrared range) by cadmium selenide quantum dots CdSe (QD) doped into the pores of anodic alumina. The authors have shown that a light absorption by the dots is enhanced by increasing the height and decreasing the pore diameter (in general, variations in height, diameter, distances between pores, etc.). Laser ablation confirmed the enhancement of light absorption by the CdSe QD. Also, the authors have discovered that the optical mode coupling of alumina and quantum dots can be improved by reducing the pore diameter and periodic distance, as well as increasing the height. According to the abstract of the paper, "the experiment has shown an improvement in laser-induced damage at 355 nm after coating 6061 aluminum alloy with functional films, which are fabricated based on numerical computing results. The title is quite adequate and appropriate. The introduction is a very useful tour d'horizon to state of art of laser-driven controlled inertial confinement fusion problem. The abstract contains the essential information of the article; The parts describing the main results are clearly and correctly written.
The paper is very actual, interesting, original and novel and adds so much to results that are already published. My understanding is that the paper by Jiaheng Yin et al is aimed for the MDPI journal “Nanomaterials” so its style is adequate to the purpose, and the length of a paper is justified by its contents.
There are a few minor points, which should be clarified in order to meet the possible questions of the readers and to make the contribution even more appealing to a wider audience (with the consent of the authors):
i). The title, as well as the abstract of the article, should be corrected. The authors should clearly explain all the abbreviations used. In the title of the article, the use of abbreviations such as AA (aluminum alloy) and other nonexplained symbols should be avoided.
ii), All the numerical results in the paper have been received “by using commercial-grade simulator based on so called “Finite differences Time Dependent FDTD method”. It would be very desirable (especially for the readers) to give at least a brief description (at the level of a few lines) and the main physical and computer provisions of this method. This will allow readers to judge both the correctness of the results in the light of the extreme complexity of the problem, and possible further generalizations and improvements of the desired results for other materials. It is also desirable to give references to the authors of the method, and also (see below) to note alternative methods for describing the interaction of materials with laser radiation).
iii), The authors have declared that “… experiments indicated that AAO pores with embedded QDs improved AA 6061 resistance to laser-induced damage with wavelength of 355 nm”. It would be desirable to give a fuller explanation of these experiments (not only in the supplementary material).
iv). The list of references could be increased (at the author’s discretion) in order to at least briefly cover the key methods and algorithms to analysis, modeling the interactions “Laser radiation- Materials” (such as Green’s Function method, S-matrix formalism, Density functional theory etc). For example, the following references can be very useful for reader (see the attached pdf-file).
v) Please, the authors should carefully proof-read the manuscript to minimize typographical, editorial or other misprints, check that all symbols (in the text, formula, tables and figures) and abbreviators are defined etc; The references list should be also checked and corrected; for example, the reference [15] should be looked as: Saman, Kohnehpoushi, Mehdi, Eskandari, Bahram, Abdollahi, Nejand, Vahid, Ahmadi, Numerical calculation of visible light absorption enhancement of CdSe-quantum dot-sensitized TiO2 nanorod periodic array as photoanode. Journal of Physics D: Applied Physics, Volume 50, Issue 7, article id. 075102 (2017). DOI: 1088/1361-6463/aa560f
Recommendation: The scientific merit of the paper is very high. The paper should be recommended for publication in the MDPI journal “Nanomaterials” provided the authors comply with the minor points listed. Thanks to authors for minor revision.
Author Response
i). The title, as well as the abstract of the article, should be corrected. The authors should clearly explain all the abbreviations used. In the title of the article, the use of abbreviations such as AA (aluminum alloy) and other nonexplained symbols should be avoided.
Response: Thank you for your careful review, and we have revised the abbreviations such as AA(aluminum alloy) and other symbols.
ii). All the numerical results in the paper have been received “by using commercial-grade simulator based on so called “Finite differences Time Dependent FDTD method”. It would be very desirable (especially for the readers) to give at least a brief description (at the level of a few lines) and the main physical and computer provisions of this method. This will allow readers to judge both the correctness of the results in the light of the extreme complexity of the problem, and possible further generalizations and improvements of the desired results for other materials. It is also desirable to give references to the authors of the method, and also (see below) to note alternative methods for describing the interaction of materials with laser radiation).
Response: Thanks for your suggestions. We have added a brief description of FDTD method, and the main physical and computer provisions of this method have been given in page 5, line 2-9. The reference to the authors of the method has been added seen in Ref 33.
iii). The authors have declared that “… experiments indicated that AAO pores with embedded QDs improved AA 6061 resistance to laser-induced damage with wavelength of 355 nm”. It would be desirable to give a fuller explanation of these experiments (not only in the supplementary material).
Response: Thanks for your reminder. We have given a fuller explanation shown in conclusion part.
iv). The list of references could be increased (at the author’s discretion) in order to at least briefly cover the key methods and algorithms to analysis, modeling the interactions “Laser radiation- Materials” (such as Green’s Function method, S-matrix formalism, Density functional theory etc). For example, the following references can be very useful for reader (see the attached pdf-file).
Response: Thanks for your advices. We have added discrete dipole approximation and Green dyadic tensor method in page 3, line 6-1 and gave its references.
v). Please, the authors should carefully proof-read the manuscript to minimize typographical, editorial or other misprints, check that all symbols (in the text, formula, tables and figures) and abbreviators are defined etc; The references list should be also checked and corrected; for example, the reference [15] should be looked as: Saman, Kohnehpoushi, Mehdi, Eskandari, Bahram, Abdollahi, Nejand, Vahid, Ahmadi, Numerical calculation of visible light absorption enhancement of CdSe-quantum dot-sensitized TiO2 nanorod periodic array as photoanode. Journal of Physics D: Applied Physics, Volume 50, Issue 7, article id. 075102 (2017). DOI: 1088/1361-6463/aa560f.
Response: Thank you for your advice. We have carefully checked all symbols and revised the reference “Saman, Kohnehpoushi, Mehdi, Eskandari, Bahram, Abdollahi, Nejand, Vahid, Ahmadi, Numerical calculation of visible light absorption enhancement of CdSe-quantum dot-sensitized TiO2 nanorod periodic array as photoanode. Journal of Physics D: Applied Physics, Volume 50, Issue 7, article id. 075102 (2017). DOI: 1088/1361-6463/aa560f”. Thank you again.
Reviewer 2 Report
The present article illustrates a potential application of quantum dots for light absorption enhancement. The material presented is suitable for publication after considering the following :
- Check grammar across the document
- The experimental part should be at the body of the document
- What instrumentation was used?
- Laser conditions?
- Sample preparation?
- Materials?
- There are some comments questions on the document (attached)

Author Response
Reviewer Comments: The present article illustrates a potential application of quantum dots for light absorption enhancement. The material presented is suitable for publication after considering the following :
1 Check grammar across the document.
Response: Thanks for your reminder, and we have checked grammar across the document and revised the unproper parts.
2 The experimental part should be at the body of the document
(1)What instrumentation was used?
(2)Laser conditions?
(3)Sample preparation?
(4)Materials?
Response: Thanks for your kind reminder, and we have added the experimental part which detailly explained the used instrumentation, laser condition, sample preparation and materials.
3 There are some comments questions on the document (attached).
(1) can the authors be more specific what message they want to convey?
Response: Thanks for your suggestion, and we have revised your marked sentence and made it be more specific.
(2) Can you comment on the Cd, Se and Si distribution on the surface?
Response: Thanks for your advice, and we have added description of Cd, Se and Si distribution on the surface on page 9 “The thickness of SiO2 film is about 200 nm, so the distribution of surface are only silicon element and oxygen element.” .